# High triplet energy host material with a 1,3,5-oxadiazine core from a one-step interrupted Fischer indolization
Charlotte Riley[1], Hwan-Hee Cho [2], Alexander C. Brannan[1], Nguyen Le Phuoc[3], Mikko Linnolahti [3] ✉, Neil C. Greenham [2] ✉ & Alexander S. Romanov [1] ✉

Energy-efficient and deep-blue organic light-emitting diode (OLED) with long operating stability remains a key challenge to enable a disruptive change in OLED display and lighting technology. Part of the challenge is associated with a very narrow choice of the robust host materials having over 3 eV triplet energy level to facilitate efficient deep-blue emission and deliver excellent performance in the OLED device. Here we show the molecular design of new 1,3,5-oxadiazines (NON)-host materials with high triplet energy over 3.2 eV, enabling deep-blue OLED devices with a peak external quantum efficiency of 21%. A series of NON-host materials are prepared by the condensation of substituted arylhydrazines and cyclohexylcarbaldehyde in a 2:3 ratio. This straightforward "one-pot" procedure enables the formation of indoline-containing derivatives with three fused heterocyclic rings and two stereogenic centres. All materials emit UV-fluorescence in the range of 315–338 nm while possessing highly desirable characteristics for application in deep-blue OLED devices: good thermal stability, a wide energy gap (3.9 eV), a high triplet energy level of (3.3 eV), and excellent volatility during sublimation.

Since the seminal publication by Tang and Van Slyke in 1987, Organic Light-Emitting Diodes (OLEDs) have emerged as a dominant technology within the display and lighting industries[1]. Significant progress in the OLED field has been driven by the development of new energy-efficient materials and OLED fabrication architectures[2]. The first generation of OLED devices utilized organic fluorescent emitters. When charge is injected into an OLED device, it results in the simultaneous generation of one singlet and three triplet excitons, as determined by spin statistics. However, fluorescent emitters limit the internal quantum efficiency (IQE) of first-generation OLEDs to 25% due to nonradiative loss of the triplet excitons[3]. In contrast, the second and third generations of OLED devices achieve up to 100% IQE by harvesting all singlet and triplet excitons. This improvement is achieved through phosphorescence and thermally activated delayed fluorescence (TADF) mechanisms[4–10].

A significant challenge in the OLED field lies in developing energy-efficient blue OLEDs with high operating stability[11]. Phosphorescent and TADF emitters often suffer from self-quenching due to triplet–triplet or triplet-polaron annihilation processes[12,13]. To mitigate this, the "solid-state dilution" approach is employed, necessitating the use of a host matrix[12,13]. Similar to emitter molecules, host materials must meet several criteria to enable energy-efficient and stable blue OLEDs. These include (1) a higher triplet ($T_1$) energy level compared with the emitter molecules to prevent

Dexter energy transfer to the host; (2) the highest occupied (HOMO) and lowest unoccupied (LUMO) molecular orbital energy levels for the host should confine those of the emitter molecule; (3) balanced charge-carrier (electron and hole) injection and mobility within the host; (4) high thermal and redox stability; (5) high volatility during the deposition process; (6) ability to enable slow roll-off of efficiency at high luminance, and many more[14]. All these requirements pose a unique challenge for blue OLED hosts, limiting the number of appropriate materials. Scheme 1 presents some of the most frequently used high triplet energy materials, illustrating various molecular design strategies used to achieve the necessary energetic properties for the blue OLED host materials[15–17]. Synthetic protocols to obtain such hosts (Scheme 1) are optimized but involve expensive reagents and lengthy multistep syntheses which frequently employ transition metal catalysts that contribute to the final material cost. Additionally, the environmental impact of producing advanced materials (Scheme 1) is increased by necessary purification steps, such as a double sublimation[18]. In this work, we introduce a novel molecular design for a blue host material containing a 1,3,5-oxadiazine core which is unprecedented in the literature. Such **NON**-type materials are synthesised in a simple 'one-pot' reaction using inexpensive reagents in a good yield. We demonstrate that this new NON-type (Scheme 1) host material is compatible with deep-blue emitters. An external

[1]Department of Chemistry, The University of Manchester, Manchester, UK. [2]Department of Physics, Cavendish Laboratory, Cambridge University, Cambridge, UK. [3]Department of Chemistry, University of Eastern Finland, Joensuu, Finland. ✉e-mail: mikko.linnolahti@uef.fi; ncg11@cam.ac.uk; alexander.romanov@manchester.ac.uk

**Scheme 1 | Molecular structures for organic host materials.** Molecular structure for popular **DPEPO**[19], **PPBi**[20], **UGH3**[42], **oCBP**[29,43] and **NON** (R = H, CF₃ or Br) OLED host materials with triplet energy level more than 3.0 eV.

**oCBP**
T₁ = 3.0 eV

**DEPEPO**
T₁ = 3.0 eV

This work:
**NON**
T₁ = 3.35 eV

**PPBi**
T₁ = 3.35 eV

**UGH3**
T₁ = 3.10 eV

**Scheme 2 | Synthetic protocols.** One-step synthesis of the compounds **NON, NON-CF₃** and **NON-Br**.

**Garg et al., 2009**

AcOH

3 h, 60°C

Yields: 59-81%

**This work:**

AcOH

3 h, 60°C

| R = H | NON |
| R = CF₃ | NON-CF₃ |
| R = Br | NON-Br |

Yields: 22-82%

quantum efficiency (EQE) up to 21% was achieved in OLEDs with a device performance on par with hosts of complex molecular design, such as DPEPO[19] or PPBi[20].

## Results and discussion

### Synthesis and characterization of NON-materials

The novel 1,3,5-oxadiazine compounds **NON, NON-CF₃** and **NON-Br** were synthesized in one step by heating the substituted arylhydrazine hydrochlorides and cyclohexylcarbaldehyde in glacial acetic acid at 60 °C for 3 h (Scheme 2). The reaction forms three heterocyclic rings and two stereogenic centres (C2 and C3 carbon atoms, Fig. 1). The reaction yields varied from moderate for **NON** and **NON-Br** (22–26%) to high for **NON-CF₃** (82%). Garg et al.[21] previously reported that lactols and hemiaminals behave as masked carbaldehydes under the conditions of a Fisher indole synthesis, resulting in the formation of various indoline derivatives with two heterocyclic rings and two stereogenic centres (Scheme 2, top). Therefore, we propose that the formation of the **NON-R** compounds occurs via an interrupted Fischer indolization[22,23]. It's worth noting that the 1,3,5-oxadiazine moiety is a powerful heterocyclic framework with applications in pharmaceuticals, agriculture, and industry. This makes the facile synthesis of **NON-R** (R = H, CF₃ and Br) materials interesting in its own right[24–28].

All NON-compounds are white solids, stable in air and soluble in common polar organic solvents dichloromethane, acetone, THF and poorly soluble in hexane. The purity of the products was confirmed by ¹H, ¹³C and ¹⁹F NMR, High-Resolution Mass Spectrometry and X-ray diffraction analysis. The thermal stability of NON-materials increases from **NON-Br** (235 °C) to **NON** (256 °C) to **NON-CF₃** (262 °C, Fig. S12). The central oxadiazine ring possesses two groups of distinctive proton resonances: the first as a doublet at

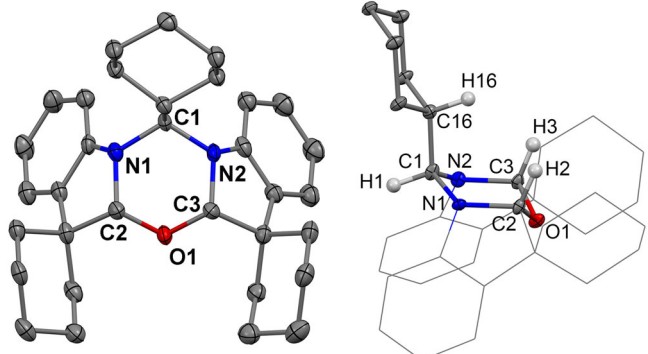

**Fig. 1 | X-ray crystal structure of NON-compound.** X-ray crystal structure of NON-compound front view (left, hydrogen atoms omitted for clarity) and side view (right, with selected hydrogen atoms). Ellipsoids are shown at the 50% level.

4.9 ppm for C1–H1, and the second is a singlet at 5.2 ppm for C2–H2 and C3–H3 (see Supplementary File). Crystals of **NON**-materials were grown by layering dichloromethane solutions with hexanes (Figs. 1, S8–S10). All three compounds possess *meso*-stereochemistry (R for C2 and S for C3). Similar stereochemical observations for the fused 1,3,5-oxadiazine rings have been reported by Ghinet et al.[24]. The X-ray crystal structure revealed that the central oxadiazine six-membered ring adopts a chair-conformation where the H2 and H3 atoms occupy an axial position, whereas the H1 atom occupies an equatorial position (Fig. 1). Both indoline and spiro-cyclohexyl moieties of the NON-materials point in opposite directions with respect to the central

**Table 1 | Cyclic voltammetry and additional photophysical parameters**

|  | Oxidation | | | $E_{HOMO}$ | $E_{LUMO}$ | $E_{opt\text{-}gap}$ | $\lambda_{abs}$ (nm),[b] | $S_1$ | $T_1$ | $S_{H/L}$[d] |
|---|---|---|---|---|---|---|---|---|---|---|
|  | $E_{1st}$ | $E_{onset\ ox}$ | $E_{2nd}$ | eV | eV[a] | eV | ($\varepsilon \times 10^3$ $M^{-1}$ $cm^{-1}$) | eV[c] | eV[c] |  |
| **NON** | +0.50 | +0.37 | – | –5.76 | –1.90 | 3.86 | 264 (28), 294 (10.5), 304 (sh) | 3.91 | 3.35 | 0.62 |
| **NON-CF₃** | +0.90 (100) | +0.75 | +1.38 | –6.14 | –2.24 | 3.9 | 273 (21.2), 302 (sh) | 4.36 | 3.31 | 0.66 |
| **NON-Br** | +0.62 (90) | +0.51 | +1.12 | –5.90 | –2.21 | 3.69 | 270 (15.7), 304 (4.1) | 4.36 | 3.21 | 0.62 |

[a]In 1,2-difluorobenzene (DFB) solution, recorded using a glassy carbon electrode, concentration 1.4 m$M$, supporting electrolyte [$nBu_4N$][$PF_6$] (0.13 $M$), measured at 0.1 V s$^{-1}$ while Ag wire was used as quasi-reference electrode. $E_{HOMO} = -(E_{onset\ ox\ Fc/Fc+} + 5.39)\ eV$; $E_{LUMO} = E_{opt\ gap} + E_{HOMO}\ eV$, see Table S2 and Fig. S11.
[b]Concentration $3.57 \times 10^{-5}$ M for **NON**, $3.16 \times 10^{-5}$ M for **NON-CF3**; $2.69 \times 10^{-5}$ M for **NON-Br**.
[c]$S_1$ and $T_1$ energy levels based on the onset values of the emission spectra blue edge in MeTHF glasses at 77 K and 298 K.
[d]$S_{H/L}$ is an overlap integral between HOMO and LUMO calculated by DFT.

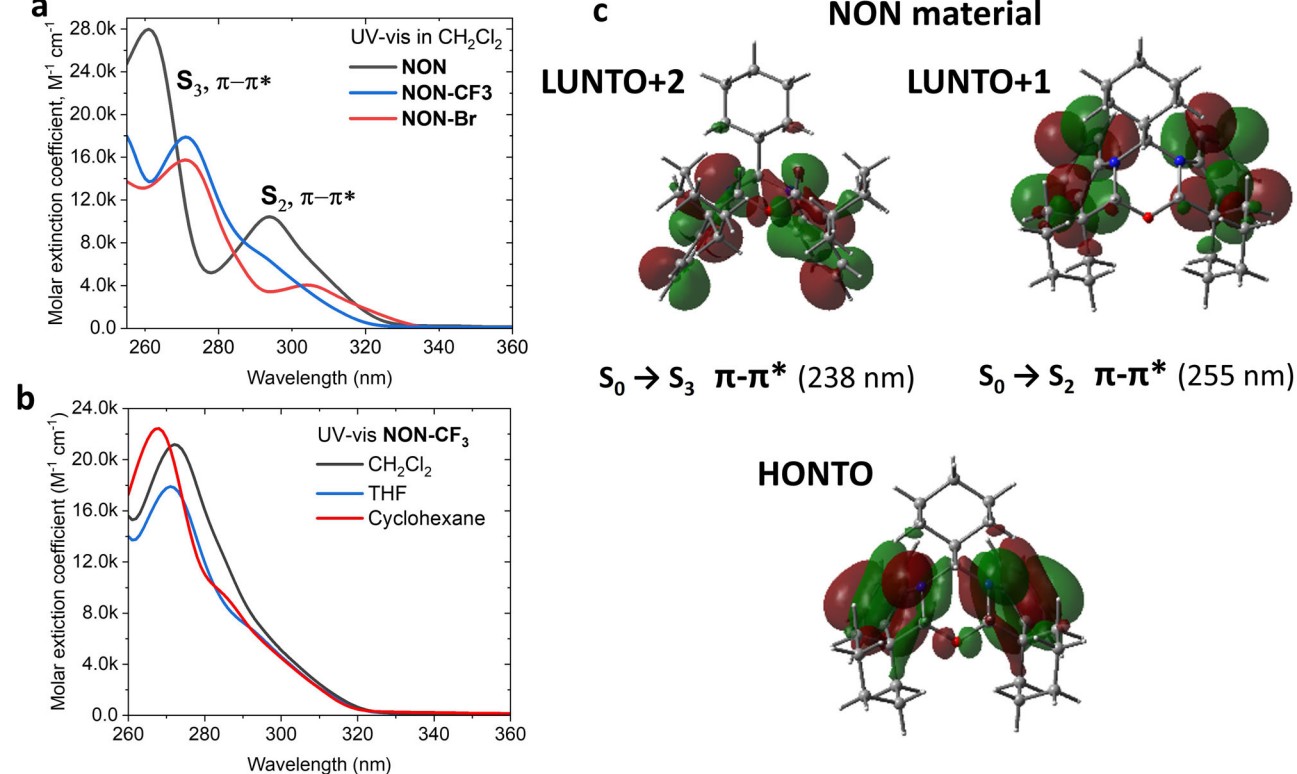

**Fig. 2 | UV–vis absorption and frontier molecular orbitals distribution for NON-materials.** UV-vis absorption for NON-materials in dichloromethane solution (**a**) and solvent of different polarity for **NON-CF₃** (**b**); molecular orbitals distribution involved in vertical excitations for NON-materials (**c**). There is ca 40 nm offset between the gas-phase calculated energy of the transition and observed UV-vis absorption in solution.

oxazine core, thus forming voids above and below the central ring, which are capable of accommodating solvent molecules (for instance, hexane, see Fig. S8). Molecules of the NON-materials form a three-dimensional network through weak C–H···π, C–H···F, C–H···Br or C–H$^{\delta+}$(phenyl)···$^{\delta-}$H–C(cyclohexyl) interactions between neighbouring molecules.

Cyclic voltammetry experiments were performed to investigate the redox properties of **NON, NON-Br**, and **NON-CF₃** in 1,2-difluorobenzene solution (Fig. S11). The data is summarised in Table 2. The NON-compound exhibits non-reversible oxidation at a peak potential of $E_p = +0.5$ V. The **NON-CF₃** material shows two oxidation processes: the first one is reversible ($E_{1/2} = +0.90$ V), while the second is irreversible with $E_p$ at +1.38 V. The **NON-Br** compound shows two reversible oxidations at $E_{1/2} = +0.62$ and +1.12 V. The onset of the first oxidation potential ($E_{onset}$) values was used to calculate the energy level of the highest occupied molecular orbital ($E_{HOMO}$, see Table 1) for the **NON-R** materials. The $E_{HOMO}$ energy correlates with the electron-withdrawing strength of the substituent **R**, i.e., it increases in line from –5.76 (R = H) to –5.90 (R = Br) and –6.14 eV (R = CF₃). The oxidation process of the NON-compounds is likely centred on the indoline moiety, which is supported by the theoretical calculations which predict the localization of the HOMO to this region (see Fig. 3c, Table S4). The reduction potential for all three compounds was beyond the solvent discharge potential, therefore, a direct calculation of the LUMO energy level is impossible. Instead, the $E_{LUMO}$ energy was estimated indirectly using the equation $E_{LUMO} = E_{opt\text{-}gap} + E_{HOMO}$, where $E_{opt\text{-}gap}$ is taken as the red-onset of the lowest energy absorption band in the UV-vis absorption spectra in a dichloromethane solution (*vide infra*, Table 1).

**Photophysical properties**
To investigate photophysical properties, UV-visible absorption (Fig. 2) and photoluminescence (Fig. 3) spectra were recorded for all NON-compounds. The data is summarized in Tables 1 and 2. All compounds show a strong absorption band in the range 260–280 nm (extinction coefficient ε up to $2.8 \times 10^4$ $M^{-1}$ $cm^{-1}$, Table S3) and a low-energy absorption band at *ca*. 304 nm with ε coefficient up to $4 \times 10^3$ $M^{-1}$ $cm^{-1}$ (Table S3). Theoretical calculations revealed that oscillator strength coefficients (*f*) for $S_0 \rightarrow S_1$ (HOMO→LUMO) are nearly zero for all NON-materials. Further analysis of the TD-DFT results (Fig. 2), for instance for the NON-compound,

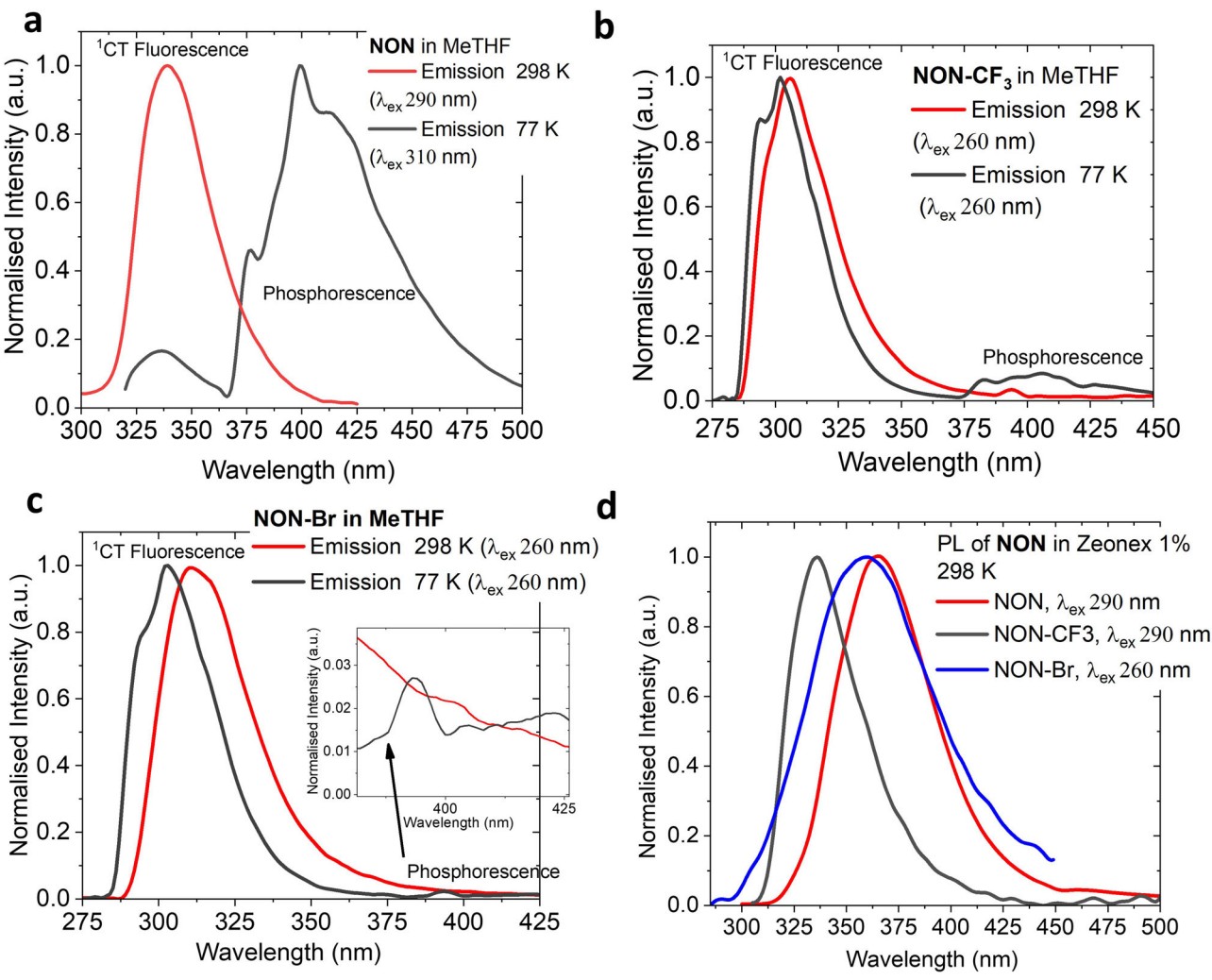

**Fig. 3 | Photoluminescence spectra for NON-materials.** Photoluminescence (PL) spectra for compounds **NON** (**a**), **NON-CF₃** (**b**) and **NON-Br** (**c**) in MeTHF solution (at 298 K and frozen glass at 77 K); PL for NON-compounds in 1% by weight Zeonex matrix at 298 K (**d**).

**Table 2 | Photophysical properties of the NON-materials in MeTHF solution and 1% Zeonex matrix at 298 and 77 K**

| | 298 K | | | $k_r$ | $k_{nr}$ | 77 K | |
|---|---|---|---|---|---|---|---|
| | $\lambda_{max}$ (nm) | $\tau$ (ns) | $\Phi$ (%, $N_2$)[a] | $(10^8 s^{-1})$[b] | $(10^8 s^{-1})$[c] | $\lambda_{max}$ (nm) | $\tau$ |
| **MeTHF solution** | | | | | | | |
| NON | 338 | 1.4 | 23 | 1.7 | 5.5 | 337<br>400 | 2.6 ns<br>>3 s |
| NON-CF₃ | 310 | 0.9[d] | 12 | 1.4 | 10 | 302<br>400 | 5.9 ns<br>>3 s |
| NON-Br | 315 | 1.1[d] | – | – | – | 302<br>393 | –<br>>3 s |
| **1% Zeonex matrix** | | | | | | | |
| NON | 336 | 1.8 | 21 | 1.2 | 4.4 | – | – |
| NON-CF₃ | 306 | 0.9[d] | 10 | 1.2 | 10 | – | – |
| NON-Br | 350 | 1.1[d] | 3 | 0.27 | 8.7 | – | – |

[a]Quantum yields determined using an integrating sphere.
[b]Radiative constant $k_r = \Phi/\tau$.
[c]nonradiative constant $k_{nr} = (1 - \Phi)/\tau$.
[d]In case of two-component lifetime $\tau$ an average was used: $\tau_{av} = (B_1/(B_1 + B_2))\tau_1 + (B_2/(B_1 + B_2))\tau_2$, where $B_1$ and $B_2$ are the relative amplitudes for $\tau_1$ and $\tau_2$, respectively.

revealed the intense vertical excitation: $S_0{\rightarrow}S_2$ ($f = 0.08$; HOMO→LUMO + 1) and $S_0{\rightarrow}S_3$ ($f = 0.17$; HOMO→LUMO + 2) which are all ascribed to π–π* transitions within the indolines of the NON-materials (Tables S4, S6 and S7). The optical gap ($E_{opt\text{-}gap}$) values for all NON-materials were calculated from the red-onset of the lowest energy absorption band: 3.72 eV for **NON**, 3.86 eV for **NON-Br** and 3.90 eV for **NON-CF₃**. All absorption bands demonstrate very little solvatochromic effect, as shown in Fig. 2b for the compound **NON-CF₃** (see Supporting Files, Figs. S13–S15 for

all compounds). This indicates only a minor change in the dipole moment upon vertical excitation from the ground $S_0$ to excited $S_1$ states and is reflected in the TD-DFT theoretical calculations (Table S5).

All NON-compounds exhibit featureless and narrow near-UV-fluorescence with a full width at half maximum (FWHM) of 3400 cm$^{-1}$ (40 nm) in MeTHF solution or Zeonex films at 295 K (Figs. 3 red line, S16–S21). The electron-withdrawing substituents on the indoline moieties lead to a blue shift in emission from 338 nm for **NON** to 310 and 315 nm for **NON-CF$_3$** and **NON-Br**, respectively. This observation is corroborated by theoretical calculations that show the HOMO is localised over the indoline moieties. Therefore, increasing the electron-withdrawing strength of the substituents attached to the indoline moiety results in the stabilization of the HOMO energy level and an increase in the emission energy. The photoluminescence quantum yields (PLQY) for UV-luminescence increase from **NON-Br** (3%) to **NON-CF$_3$** (12%) to **NON** (23%) in the MeTHF solution at 295 K.

All NON-materials, in all media, exhibit an excited state lifetime on the nanosecond time scale in the range of 0.9–1.8 ns, which is typical for the fluorescence emission mechanism. Figure 3 depicts the emission profiles of the NON-compounds in frozen MeTHF glasses at 77 K (black line, Table 2). The emission spectra of **NON** at 77 K is dominated by a deep-blue and vibronically resolved phosphorescence, peaking at 400 nm. The excited state lifetime is too long to be measured by a gated Xenon flashlamp. However, the emission was observed with the naked eye to persist for more than 3 s after the excitation ceased. Unlike **NON**, the emission spectra of **NON-CF$_3$** and **NON-Br** compounds are dominated by UV-fluorescence (306–337 nm), with a minor contribution from the deep-blue phosphorescence component at 77 K (Fig. 3b, c). Similar to **NON**, the excited state lifetime of the phosphorescence is longer than 3 s, while the fluorescence excited state lifetime only slightly increases up to 5.9 ns upon cooling.

The energy levels of the singlet ($S_1$) and triplet ($T_1$) excited states for all NON-materials were deduced from the blue-onsets of the fluorescent and phosphorescent profiles (Figs. 3, S23, Table 1) at 77 K. The **NON** and **NON-CF$_3$** materials exhibit a high triplet energy level at 3.3 eV, which is higher than the most popular host material DPEPO (3.0 eV) and on par with the most stable deep-blue OLED host material – PPBi (3.3 eV). The computed values align well with the experimental results (Table S6). The theoretical calculations show that both frontier orbitals, HOMO and LUMO, are primarily localized over the indoline rings of the NON-compounds (Table S4), resulting in a large frontier orbital overlap integral (0.62–0.66) for all NON-materials (Table S4). This is largely due to the absence of spatially separated donor and acceptor moieties, which results in a significant stabilization of the $T_1$ energy level and an energy difference with the singlet excited state, $\Delta E_{ST}$, up to 1.15 eV. Note that the central oxadiazine core with spiro-sp$^3$ carbon atoms acts as an insulator, preventing extended conjugation between two indoline moieties for the NON-materials, which prevents further lowering of the $T_1$ energy level. It has previously been demonstrated that small molecule hosts[29,30] show triplet energy variance depending on the degree of conjugation and magnitude of the dihedral angle between the cycles involved, for instance, carbazole moieties in CBP ($T_1$ = 2.64 eV), mCBP ($T_1$ = 2.84 eV), oCBP ($T_1$ = 3.0 eV, Scheme 1)[29] and others[29,30]. Therefore, the novelty of NON-host molecular design strongly benefits from the bulky cyclohexyl moieties, absence of any inter-cycle bonds and spiro-sp$^3$ carbon atom breaking the conjugation thus explaining the high triplet energy level and making NON-hosts more robust towards the local environment.

### Device fabrication, characterisation, and performance

We assessed all NON-materials against the criteria outlined in the introduction to identify a promising host material for the development of deep-blue OLED devices. The **NON-CF$_3$** material exhibits good thermal stability, a wide energy gap (3.9 eV) thanks to a well-stabilised HOMO (–6.1 eV) and a destabilised LUMO (–2.2 eV), a high triplet energy level of 3.3 eV, and excellent volatility during the sublimation process due to three cyclohexyl groups. These attributes motivated us to test the **NON-CF$_3$** material as a host with our recently developed Carbene-Metal-Amide (CMA) TADF material **P170** (Fig. 4a)[31]. First two triplet states of complex **P170** are $T_1$ ($^3$CT 2.77 eV) and $T_2$ ($^3$LE

3.29 eV)[31] are lower in energy compared to the first triplet state of the **NON-CF$_3$** host ($T_1$ is 3.3 eV) thus preventing the Dexter energy transfer back to the host material. OLED devices were fabricated via thermal vapour deposition. The device architecture is shown in Fig. 4, along with the chemical structures of the materials used. The 30 nm layer of 1,1,-bis{4-[N,N-di(4-toyl)amino]phenyl}cyclohexane (TAPC) and a 10 nm layer of 9,9′-biphenyl-2,2′-diylbis-9H-carbazole (oCBP) function as hole transport layers. In each device, the 20 nm thick emissive layer (EML) consists of an emitting material doped at 10% by weight into the **NON-CF$_3$** host. A 45 nm layer of diphenyl-4-triphenylsilyl-phenylphosphine oxide (TSPO1) is used for electron transport. Four EML compositions were prepared using four different host materials, namely PPBi, TCP, mCP and **NON-CF$_3$** (Fig. 4a), to evaluate and compare the performance of **NON-CF$_3$**.

The electroluminescence spectra for devices at 6 V are shown in Fig. 4, and a summary of the performance data for each device is given in Table 3. The use of **NON-CF$_3$** as a host material results in a blue shift of up to 31 nm for the electroluminescence from **P170** compared to the other host materials tested. This leads to a highly desirable 448 nm EL from **P170** with a narrow FWHM of ca. 60 nm in a **NON-CF$_3$** host. This result is favourable when comparing **NON-CF$_3$** to the DPEPO host, where the deep-blue OLED (EML: **P170** 10% in DPEPO) demonstrated a 30 nm wider FWHM value (ca. 90 nm)[31]. Therefore, a commonly cited disadvantage of organometallic TADF materials, such as their broad EL emission profiles associated with the emission occurring from a charge transfer (CT) state, could be circumvented by using the **NON-CF$_3$** host material. This enables narrow EL profiles and better CIE-colour coordinates. The phenomena of the PL[31–34] and EL[31,35] blue shift of the CT emission for CMA materials were studied in detail and explained by thermally activated diffusion and electrostatic interactions between CMA emitter with host molecules.

Devices utilizing **NON-CF$_3$** exhibit an increased turn-on voltage, which is reported as the applied bias for luminescence to equal 1 cd m$^{-2}$. This suggests a higher barrier for charge injection into the emitting layer. Figure 4 shows the current density–voltage–luminescence (J–V–L) curves for each OLED device. Devices experience luminescence roll-off when operating above 11 V, indicating a need for further improvement in the molecular design of the NON-host materials. However, the OLED devices hosted by **NON-CF$_3$** achieved an External Quantum Efficiency EQE$_{max}$ value of 21.0%, the highest value among all tested host materials. Impressively, OLED devices containing P170 doped into **NON-CF$_3$** were operational at 100 cd m$^{-2}$ with an EQE of 9.2%, which is comparable to commercially available host materials such as TCP and mCP tested in this work. The EQE performance is likely associated with higher PLQY values for **P170** films in the **NON-CF3** host (82%) compared to TCP films (38%). These findings suggest a high potential for NON-host materials to be used as host materials for deep-blue OLEDs.

### Conclusion

We discovered an unexpected product of the interrupted Fischer indolization reaction which has enabled a straightforward synthesis of optionally substituted 1,3,5-oxadiazines (NON) in good yields. Our protocol involves a single-step condensation reaction between cost-effective substituted hydrazines and carbaldehydes to prepare NON-compounds with good thermal stability up to 262.5 °C. We demonstrated that all NON-materials emit only fast UV-fluorescence (310–338 nm) at room temperature, while a long-lived (over 3 s) deep-blue phosphorescence emerges upon cooling to 77 K. Our theoretical calculations revealed large frontier orbitals overlap integral values up to 0.66 for the NON-materials which is rationalised by the significant energy difference between singlet ($S_1$) and triplet ($T_1$) excited states up to 1.15 eV.

Our photophysical and electrochemical investigation into the new 1,3,5-oxadiazine materials revealed their suitability, from an energy level standpoint, for use as host materials in OLED devices. **NON-CF$_3$** was selected for use in proof-of-concept devices based on its good thermal stability, wide energy gap (3.9 eV), well-stabilised HOMO (–6.1 eV) and destabilised LUMO (–2.2 eV), high triplet energy level (3.3 eV), and excellent volatility, thanks to the bulky cyclohexyl groups that prevent strong intermolecular contacts. Given the novel molecular design of NONs and their remarkably

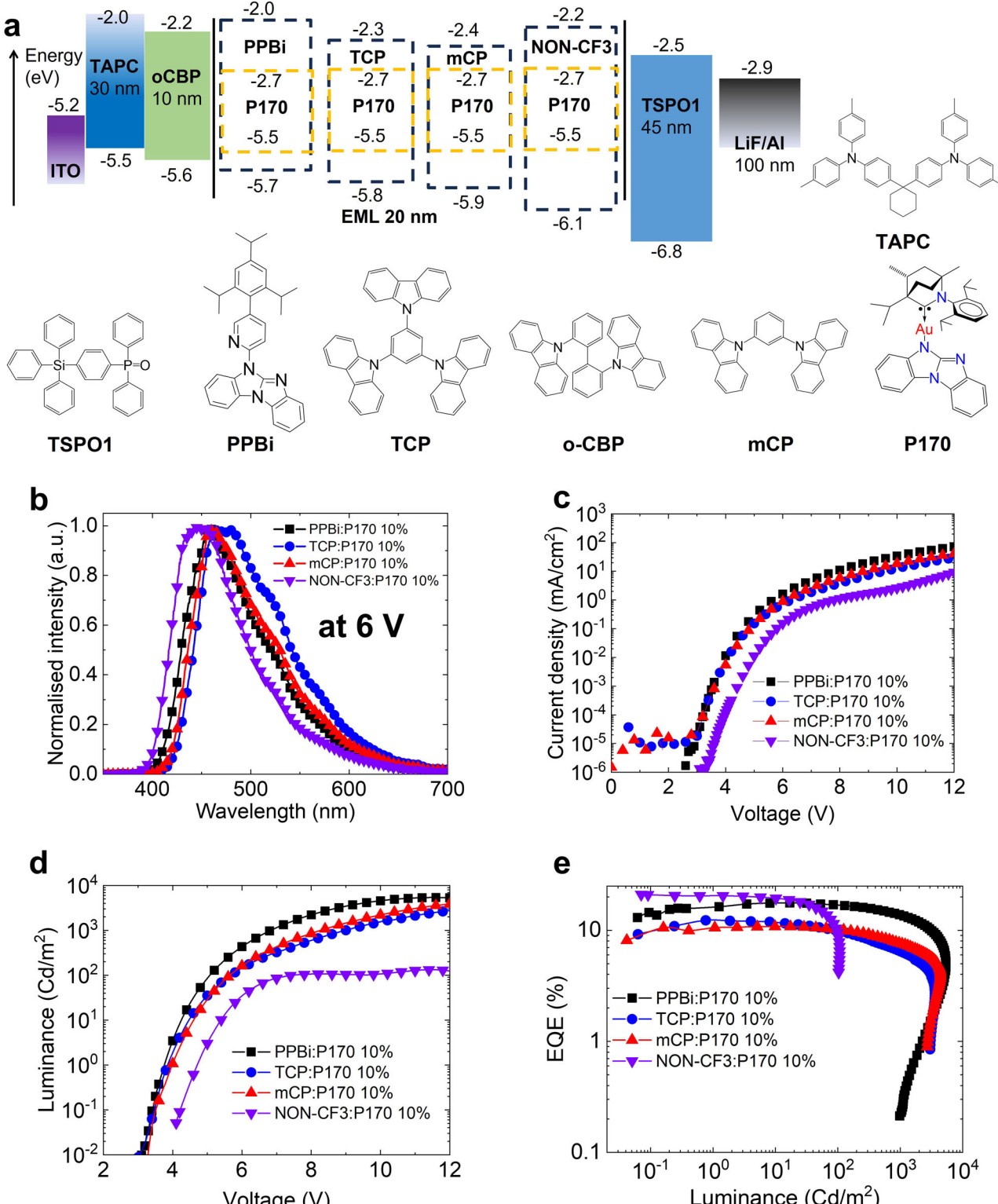

**Fig. 4 | Vapour-deposited OLED device architectures. a** OLED architecture for **P170** doped into different hosts at 10% by weight and chemical formulae of materials used in the devices; **b** Normalised electroluminescence spectra from devices incorporating **P170** 10% by weight in host-guest structures; **c** current density-voltage plot; **d** luminance-voltage plot; **e** EQE versus luminance plot.

simple synthesis, we are impressed that OLED devices using a CMA TADF emitter and **NON-CF₃** host achieved EQE values up to 21%. Furthermore, we demonstrated how the use of a NON-host material can facilitate excellent deep-blue colour emission by producing a blue-shifted and narrow (60 nm) electroluminescence profile in comparison to conventional host materials.

This work introduces a new class of OLED host materials. Further modifications to the host material structure can be made by decorating with electron donating or withdrawing groups to enhance hole and electron transport mobility and pave the way to high-performance and stable deep-blue OLEDs.

**Table 3 | Performance data of evaporated OLEDs**

| OLED | Host | $V_{on}^a$ (V) | EQE$_{Max}$ (%) | EQE$_{100}^b$ (%) | $\lambda_{max}$ (nm) | CIE (x;y) |
|---|---|---|---|---|---|---|
| **Au1** | **1** PPBi | 3.4 | 17.7 | 16.9 | 457 | 0.18; 0.23 |
| | **2** TCP | 3.5 | 12.3 | 9.9 | 479 | 0.20; 0.30 |
| | **3** mCP | 3.5 | 10.8 | 10.2 | 461 | 0.19; 0.26 |
| | **4** NON-CF$_3$ | 4.2 | 21.0 | 9.2 | 448 | 0.17; 0.16 |

$^a V_{on}$ at 0.1 cd m$^{-2}$.
$^b$EQE at 100 cd m$^{-2}$.

## Methods

### Material preparation

The Au1 complex P170 was obtained according to the literature. All **NON**-materials were purified on a silica gel column, recrystallised from hexane, while **NON-CF3** was sublimed to have high purity for OLED fabrication (See Supplementary Methods).

### Photophysical properties

Solution UV-visible absorption spectra were recorded on a Cary 500 UV-vis-NIR spectrometer for a wavelength range 250–700 nm. Photoluminescence measurements were recorded using an Edinburgh Instruments FLS920 spectrometer. All excited state lifetimes were measured on FLS920 spectrometer with mono- and biexponential fitting provided by Edinburgh Instruments Fluoracle software v2.6.1. Absolute photoluminescent quantum yields were measured directly with a Quantaurus-QY Absolute PL quantum yield spectrometer.

### Computational details

The ground and excited states of all complexes were studied by DFT and TD-DFT[36] respectively, using Tamm-Dancoff approximation[37]. All calculations were carried out by Gaussian 16[38], employing the global hybrid MN15 functional[39] and the def2-TZVP basis set[40]. The overlap integrals between HOMO and LUMO were calculated using the Multiwfn program[41].

### OLED fabrication and characterisation

OLED devices were fabricated by high-vacuum ($10^{-7}$ Torr) thermal evaporation on ITO-coated glass substrates with sheet resistance of 15 Ω/□. Substrates were cleaned by sonication in a non-ionic detergent, deionised water, acetone, and isopropyl alcohol and subject to an oxygen plasma treatment for 10 min. Layers were deposited at rates of 0.1–2 Ås$^{-1}$. TCP, PPBi and mCP were purchased from Luminescence Technology Corp. TPBi, DPEPO and TSPO1 were purchased from Shine Materials. All purchased materials were used as received. OLED current density-voltage measurements were made using a Keithley 2400 source-meter unit. The luminance was measured on-axis using a 1 cm$^2$ calibrated silicon photodiode at a distance of 15 cm from the front face of the OLED. Electroluminescence spectra were measured using a calibrated OceanOptics Flame spectrometer. Lifetime measurements were measured with a Keithley 2400 source-meter unit and a 0.75 cm$^2$ silicon photodiode. The devices were held under rough vacuum (~$10^{-3}$ Torr).

### Data availability

Supplementary data are available in the online version of the paper. This includes Supplementary data 1, which includes the crystallographic data for this paper. The X-ray crystallographic coordinates for structures reported in this Article have been deposited at the Cambridge Crystallographic Data Centre (CCDC), under deposition number CCDC 2363678 for **NON**; 2363679 for **NON-CF₃**; 2363680 for **NON-Br**. These data can be obtained free of charge from The Cambridge Crystallographic Data Centre via www.ccdc.cam.ac.uk/data_request/cif. Authors can confirm that all relevant data are included in the paper and/ or its supplementary information files.

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

## Acknowledgements

A.S.R. acknowledges the support from the Royal Society (grant nos. URF \R1\180288, RGF\EA\181008, URF\R\231014), EPSRC (grant code EP/K039547/1). M.L. acknowledges the Academy of Finland Flagship Programme, Photonics Research and Innovation (PREIN), decision 320166, and the Finnish Grid and Cloud Infrastructure resources (urn:nbn:fi:research-infras-2016072533). N.L.P. acknowledges the Doctoral Programme in Science, Forestry and Technology (Lumeto, University of Eastern Finland). European Union's Horizon 2020 research and innovation programme grant agreement no. 101020167 (H.-H.C.). We thank Dr Louise Natrajan, EPSRC and the University of Manchester for access the Centre for Radiochemistry Research National Nuclear User's Facility (NNUF, EP/T011289/1) to use the FLS-1000 fluorometer.

## Author contributions

C.R. performed the synthesis, steady-state photoluminescence and electro-chemistry studies. H.-H.C. developed and characterized the OLED devices. A.S.R. performed X-ray crystallography. A.C.B. performed the synthesis of the **P170** complex. N.L.P. and M.L. carried out theoretical calculations. N.G. and A.S.R. planned the project and designed the experiments. C.R. and A.S.R. co-wrote the manuscript. All authors contributed to the discussion of the results, and analysis of the data and reviewed the manuscript.

## Competing interests

The authors declare no competing interests.
