## [Transparent Peer Review file · Communications Chemistry]

High Triplet Energy Host Material with a 1,3,5-Oxadiazine Core from a One-step Interrupted Fischer Indolization

Corresponding Author: Dr Alexander Romanov

Version 0:

Reviewer comments:

Reviewer #1

(Remarks to the Author)

A series of new 1,3,5-oxadiazines (NON) has been prepared. These NON materials exhibited good thermal stability, a wide energy gap, a well-stabilized HOMO, a destabilized LUMO, a high triplet energy level (3.3 eV), and excellent volatility during sublimation. The NON material was tested as a host for a Carbene-Metal-Amide complex in an OLED device, where it improved the OLED color purity of the charge transfer TADF emitter compared to devices with conventional hosts, showing a peak external quantum efficiency (EQE) of 21%. Therefore, I recommend publishing this work in Communications Chemistry after addressing the following comments, which could further enhance the manuscript's quality.

1. The authors do not explain the novelty of the material design for NON, NON-CF₃, and NON-Br. They simply state that these are novel heterocyclic frameworks. To emphasize the novelty, the authors should explain why these structured molecules are superior to known high T₁ host materials.
2. The characterization of material photophysical properties and OLED performance is brief. The CIE coordinates, EQE-luminance, and PE-luminance curves should be included in the text.
3. The explanation for the improved EQE is insufficient. Is there evidence to support the enhancement of EQE? For instance, comparing the PL quantum yields of host-guest films could be informative.
4. The operational stability results of the OLED should be provided. (This information is crucial for assessing the practical applicability of the materials.)
5. The authors should explain the significant EQE drop at 100 cd/m² (Figure 4, e).
6. Can phosphorescence be measured at 77 K without superimposed fluorescence signals by providing some delay in the timescale?
7. The authors should provide the charge transporting abilities compared with the reference compound CBP to offer a better understanding of the characteristics of the NON-host.
8. The authors synthesized the target molecules in moderate to good yields in only glacial Acetic acid condition. To increase the yields, do the authors optimize the reaction condition? If not why?

Reviewer #2

(Remarks to the Author)

The authors have presented a new candidate host material for OLEDs with a number of appealing characteristics. On the whole the manuscript is written clearly and I think it suitable for acceptance after consideration of the following comments.

- 1) A matter of taste possibly, but the authors might consider changing • for × when showing powers. I think more readers will be familiar with the latter.
- 2) The range of R groups should be defined in Chart 1.
- 3) I think that a strength of this host which has not been emphasised by the authors is the absence of any inter-cycle bonds. It has previously been demonstrated that oligomers of heterocycles show triplet energy variance depending on the magnitude of the dihedral angle between the cycles involved (see Chem Eur J, 2021, 6545). The NON system is likely much more robust towards its local environment. The authors may wish to consider this as an additional benefit of their system.
- 4) The authors should be clear about how the HOMO energy is calculated in the paper itself (maybe in the caption for Table 1) and not just the SI. On this note, the use of -5.39 for the HOMO energy of Fc is probably one of the more extreme values and is very rarely encountered. Typically groups use either -4.80 eV or -5.10 eV but the latter is much closer to observed

experimental parameters. I strongly recommend the authors re-evaluate the HOMO energy using -5.10 eV for the ferrocene HOMO. The (excellent) Adv Mater review cited leans towards the -5.10 eV. I think this change will help others to make more immediate use of the results presented.

5) In Table 1, the authors should state that a Ag wire quasi-reference was used.

6) A question born out of curiosity (the authors are welcome to ignore this) were any solubility issues encountered with difluorobenzene when running CVs? Dichlorobenzene is occasionally used but common electrolyte solubility is often poor so waveforms are rarely ideal.

7) The authors might want to double check their listed affiliations.

Reviewer #3

(Remarks to the Author)

The authors report a series of wide-bandgap host materials for blue-emitting OLEDs. The synthesis is simple but noble, systematic characterizations were carried out, and a deep-blue OLED was fabricated. This topic is interesting and timely. However, several discussions should be addressed prior to publication.

Major

Correlating electronic transitions from experimental and DFT results is too ambiguous.

The authors claim the absorption bands at 268-280 nm are S1 transitions with pi-pi transition character.

The oscillator strengths for S1 transitions are calculated to be near zero while the overlap integrals and transition dipole moments are quite large. The experimental results also show large molar absorptivity over 1e4. Generally, a larger overlap integral can contribute to a larger transition dipole moment. Consequently, this can lead to a higher oscillator strength, indicating a stronger transition probability.

T1 and T2 transitions from DFT are correlated to the absorption bands around 304 nm. However, S0-Tn transitions are not generally observed from UV-Vis. absorption spectroscopy.

Electronic transitions from DFT must be carefully analyzed one by one in order of energy with participating MOs and compared with the experimental results.

What are the meanings of PLQY, PL lifetime, and rate constant in this paper?

The rate constants are never mentioned in the main text. If not mentioned in the text, do not include them in a figure or table.

The authors should provide the PL decay curves, IRF, fitted curves, and residual analyses.

The authors claim the thermal stability of NON-CF3. It should be supported by experimental results.

Bandgap and/or T1 (, T2, T3..) energies of P170 should be compared with those of NON-CF3.

Why do EL spectra change with different host materials?

Minor

The MO diagrams are difficult to distinguish. Using a larger isodensity value is encouraged to regenerate the MO contours. Viewing from the top may also enhance visibility.

S is the common symbol for the overlap integral, not O.

Table 1 annotation: $E_{LUMO} = E_{opt\ gap} - E_{HOMO} \rightarrow E_{LUMO} = E_{opt\ gap} + E_{HOMO}$

Figure 3, Table 2 and the corresponding texts do not match. ie. temperature, PLQY of NON-Br at RT solution, etc..

i.e. Solvent, concentration should be included in the figure captions. The figures should stand alone.

Energy diagram is required for the materials used in the OLED fabrication.

Version 1:

Reviewer comments:

Reviewer #1

(Remarks to the Author)

All reviewers' comments have been addressed, and the paper is now ready for publication.

Reviewer #2

(Remarks to the Author)

The authors have addressed my points clearly. I have nothing further to add.

Reviewer #3

(Remarks to the Author)

The manuscript has been well revised according to the reviewers' previous comments. Only a few points can be discussed before publication in Communications Chemistry.

Previous comment: The authors should provide the PL decay curves, IRF, fitted curves, and residual analyses.

Response: Agreed and added Figure S22 in Supporting Information.

Table 2 and Fig S22: the RTPL decay curves are too close to the IRF. In that case, the calculated lifetime might be dominated by the IRF. Fitted curves and residual analyses with χ^2 is still missing.

Previous comment: S is the common symbol for the overlap integral, not O.

Response: We believe that it's a community dependent abbreviation while we more commonly encounter symbol "O" for the overlap integral.

S (or S_{ij}) is the widely accepted and commonly used symbol for the overlap integral in quantum chemistry. O is sometimes used for ease of notation; however, S is the standard symbol for the overlap integral. I still recommend the authors use S to avoid confusion.

Reviewer #1 (Remarks to the Author):

A series of new 1,3,5-oxadiazines (NON) has been prepared. These NON materials exhibited good thermal stability, a wide energy gap, a well-stabilized HOMO, a destabilized LUMO, a high triplet energy level (3.3 eV), and excellent volatility during sublimation. The NON material was tested as a host for a Carbene-Metal-Amide complex in an OLED device, where it improved the OLED color purity of the charge transfer TADF emitter compared to devices with conventional hosts, showing a peak external quantum efficiency (EQE) of 21%. Therefore, I recommend publishing this work in Communications Chemistry after addressing the following comments, which could further enhance the manuscript's quality.

Response: We thank referee 1 for the expert opinion.

1.The authors do not explain the novelty of the material design for NON, NON-CF₃, and NON-Br. They simply state that these are novel heterocyclic frameworks. To emphasize the novelty, the authors should explain why these structured molecules are superior to known high T₁ host materials.

Response: Many thanks for this comment. This resonates well with the first comment on molecular design from referee 2. We agree and added additional text and ref 21a,b,c as follows:

“In this work, we introduce a novel molecular design for a blue host material containing a 1,3,5-oxadiazine core which is unprecedented in the literature.” and

“It has previously been demonstrated that small molecule hosts^{21a,c} show triplet energy variance depending on the degree of conjugation and magnitude of the dihedral angle between the cycles involved, for instance, carbazole moieties in CBP (T₁ = 2.64 eV), mCBP (T₁ = 2.84 eV), oCBP (T₁ = 3.0 eV, Chart 1)^{21a} and others.^{21c} Therefore, the novelty of NON-host molecular design strongly benefit from the bulky cyclohexyl moieties, absence of any inter-cycle bonds and spiro-sp³ carbon atom breaking the conjugation thus explaining the high triplet energy level and making NON-hosts more robust towards the local environment.”

2.The characterization of material photophysical properties and OLED performance is brief. The CIE coordinates, EQE-luminance, and PE-luminance curves should be included in the text.

Response: We agree and added CIE coordinates to the Table 1, while EQE-luminance, and PE-luminance are included in the Figure 4.

3. The explanation for the improved EQE is insufficient. Is there evidence to support the enhancement of EQE? For instance, comparing the PL quantum yields of host-guest films could be informative.

Response: We agree that the difference in EQE performance is likely associated with high PLQY values in NON-host where P170 in NON-CF3 host shows PLQY up to 82% compared to 38% in TCP.

Added text: “The EQE performance is likely associated with higher PLQY values for **P170** films in **NON-CF3** host (82%) compared to TCP films (38%). These findings suggest a high potential for NON-host materials to be used as host materials for deep-blue OLEDs.”

4.The operational stability results of the OLED should be provided. (This information is crucial for assessing the practical applicability of the materials.)

5.The authors should explain the significant EQE drop at 100 cd/m² (Figure 4, e).

Response: We apologize, we are limited in capability due to problems with the thermal evaporator which is taking longer than we expected. We do not expect longer than 1h LT50 operating stability compared to our recent report for P170 in DPEPO host in *Adv. Mater.* 2404357 (2024). We are preparing a new manuscript that will explain the CMA OLED operating stability in the context of the molecular design of the CMA materials and OLED stack. We used this opportunity to demonstrate the facile synthesis of the novel NON-type host materials having a high triplet energy level. We agree that rather significant drop in EQE at 100 nits will require future modifications to enable more stable OLED device stack.

6.Can phosphorescence be measured at 77 K without superimposed fluorescence signals by providing some delay in the timescale?

Response: Yes, we agree and performed additional experiments. Added Figure S23 to the ESI. We clearly see fluorescence and phosphorescence contribution after 100 ms delay. We see no change for S1 and T1 state energy compared to our earlier results.

Figure S23. Steady state and PL profiles after various delays at 77 K for MeTHF frozen glass of compounds NON (a) and NON-CF3 (b) showing clear fluorescence and phosphorescence contribution.

7. The authors should provide the charge transporting abilities compared with the reference compound CBP to offer a better understanding of the characteristics of the NON-host.

Response: We apologize we are limited in capability of the direct electron and hole mobility measurement due to problems with the thermal evaporator

which is taking longer than we expected. However, we believe that the transporting abilities of NON-CF3 can be explained based on full device results showing higher turn-on, shallower J-V characteristics and energy levels. Overall, NON-CF3 likely to have a slower hole and electron transporting properties compared to TCP and mCP.

8. The authors are synthesized the target molecules are in moderate to good yields in only glacial Acetic acid condition. To increase the yields, do the authors are optimize the reaction condition? If not why?

Response: We attempted to use addition of Lewis acids such as ZnCl₂, however, the yields are only getting lower. The efficiency of cyclization strongly depends on the acid source and temperature. Our results in strong agreement with earlier works of Garg et al (Ref 22. *Org. Lett.*, **11(15)**, 3458, (2009)), where the acetic acid and 60C are the optimal condition to have a decent yield of the product. We consider that the 82% yield for NON-CF3 the best performing material is sufficiently high.

Reviewer #2 (Remarks to the Author):

The authors have presented a new candidate host material for OLEDs with a number of appealing characteristics. On the whole the manuscript is written clearly and I think it suitable acceptance after consideration of the following comments.

Response: We thank referee 2 for the expert opinion and corrections.

1) A matter of taste possibly, but the authors might consider changing • for × when showing powers. I think more readers will be familiar with the latter.

Response: Agreed and corrected.

2) The range of R groups should be defined in Chart 1.

Response: Agreed and corrected caption of Chart 1.

3) I think that a strength of this host which has not been emphasised by the authors is the absence of any inter-cycle bonds. It has previously been demonstrated that oligomers of heterocycles show triplet energy variance depending on the magnitude of the dihedral angle between the cycles involved (see Chem Eur J, 2021, 6545). The NON system is likely much more robust towards its local environment. The authors may wish to consider this as an additional benefit of their system.

Response: Many thanks for this comment. This resonates well with the first

comment on molecular design from referee 2. We agree and added additional text and ref 21a,b,c as follows:

“In this work, we introduce a novel molecular design for a blue host material containing a 1,3,5-oxadiazine core which is unprecedented in the literature.”
and

“It has previously been demonstrated that small molecule hosts^{21a,c} show triplet energy variance depending on the degree of conjugation and magnitude of the dihedral angle between the cycles involved, for instance, carbazole moieties in CBP ($T_1 = 2.64$ eV), mCBP ($T_1 = 2.84$ eV), oCBP ($T_1 = 3.0$ eV, Chart 1)^{21a} and others.^{21c} Therefore, the novelty of NON-host molecular design strongly benefit from the bulky cyclohexyl moieties, absence of any inter-cycle bonds and spiro- sp^3 carbon atom breaking the conjugation thus explaining the high triplet energy level and making NON-hosts more robust towards the local environment.”

4) The authors should be clear about how the HOMO energy is calculated in the paper itself (maybe in the caption for Table 1) and not just the SI. On this note, the use of -5.39 for the HOMO energy of Fc is probably one of the more extreme values and is very rarely encountered. Typically groups use either -4.80 eV or -5.10 eV but the latter is much closer to observed experimental parameters. I strongly recommend the authors re-evaluate the HOMO energy using -5.10 eV for the ferrocene HOMO. The (excellent) Adv Mater review cited leans towards the -5.10 eV. I think this change will help others to make more immediate use of the results presented.

Response: Yes, we agree that there is quite some different correction values appeared in the literature for calculating HOMO and LUMO from the CV experiment: -4.8, -5.1 and -5.39 eV. We prefer to use -5.39 eV based on our previous experiments with ultraviolet and inverse photoelectron spectroscopy to determine the exact values of HOMO and LUMO energies for the CMA materials. We found that the correction value of -5.39 eV results in the best fit between CV and photoelectron spectroscopy data. We would prefer to keep the current values for consistency with our own previous results while we provide the reference to indicate the correction value in the manuscript and Supporting Information as noted by referee.

5) In Table 1, the authors should state that a Ag wire quasi-reference was used.

Response: Agreed and added to Table 1.

6) A question born out of curiosity (the authors are welcome to ignore this) were any solubility issues encountered with difluorobenzene when running CVs? Dichlorobenzene is occasionally used but common electrolyte solubility is often poor so waveforms are rarely ideal.

Response: Yes, we prefer to use 1,2-difluorobenzene thanks to its high polarity and high solvation of various materials, wide working window, higher chemical stability (vs dichloromethane), maintain consistency of the CV conditions and comparability with our previous measurements.

7) The authors might want to double check their listed affiliations.

Response: Many thanks for the comment. Agreed and corrected

Reviewer #3 (Remarks to the Author):

The authors report a series of wide-bandgap host materials for blue-emitting OLEDs. The synthesis is simple but noble, systematic characterizations were carried out, and a deep-blue OLED was fabricated. This topic is interesting and timely. However, several discussions should be addressed prior to publication.

Response: We thank referee 3 for the expert opinion and corrections.

Major

Correlating electronic transitions from experimental and DFT results is too ambiguous. The authors claim the absorption bands at 268-280 nm are S1 transitions with pi-pi transition character. The oscillator strengths for S1 transitions are calculated to be near zero while the overlap integrals and transition dipole moments are quite large. The experimental results also show large molar absorptivity over $1e4$. Generally, a larger overlap integral can contribute to a larger transition dipole moment. Consequently, this can lead to a higher oscillator strength, indicating a stronger transition probability. T1 and T2 transitions from DFT are correlated to the absorption bands around 304 nm. However, S0-Tn transitions are not generally observed from UV-Vis. absorption spectroscopy. Electronic transitions from DFT must be carefully analyzed one by one in order of energy with participating MOs and compared with the experimental results.

Response: We agree and thank referee for this comment and correction. More vertical excitations (S1-S5 and T1-T5) were added into the Table S6 and S7. We agree the TDDFT results indicate that the oscillator strength (f) for the S0→S1 transition (HOMO→LUMO) is nearly zero for all NON-compounds. We reevaluated and made further analysis to identify transitions with high oscillator strengths (0.08 to 0.26) are S0→S2 transition (f = 0.08; HOMO→LUMO+1) and S0→S3 transition (f = 0.26; HOMO→LUMO+2). Although there is minor discrepancy between gas-phase calculations and experimental UV-vis in CH₂Cl₂, the approximate 2:1 ratio between oscillator strength coefficients for transitions to S3 and S2 states is in line with the two intense absorptions and ration between molecular extinction coefficients. All calculated structures and molecular orbital isosurfaces for S0, S2 and S3 states clearly indicate a pi-pi*-character of the

transition will slightly different alteration in involved molecular orbitals (Figure 2). We corrected Figure 2 and text accordingly.

Added text: “All compounds show a strong absorption band in the range 260–280 nm (extinction coefficient ϵ up to $2.8 \times 10^4 \text{ M}^{-1} \text{ cm}^{-1}$, Table S3) and a low-energy absorption band at *ca.* 304 nm with ϵ coefficient up to $4 \times 10^3 \text{ M}^{-1} \text{ cm}^{-1}$ (Table S3). Theoretical calculations revealed that oscillator strength coefficients (f) for $S_0 \rightarrow S_1$ (HOMO \rightarrow LUMO) are nearly zero for all NON materials. Further analysis of the TDDFT results (Figure 2), for instance for the NON-compound, revealed the intense vertical excitation: $S_0 \rightarrow S_2$ ($f = 0.08$; HOMO \rightarrow LUMO+1) and $S_0 \rightarrow S_3$ ($f = 0.17$; HOMO \rightarrow LUMO+2) which are all ascribed to π - π^* transitions within the indolines of the NON materials (Tables S4, S6 and S7). ... There is a 40 nm offset between the gas-phase calculated energy of the transition and observed UV-vis absorption in solution.”

Figure 2:

What are the meanings of PLQY, PL lifetime, and rate constant in this paper? The rate constants are never mentioned in the main text. If not mentioned in the text, do not include them in a figure or table.

Response: PLQY and PL lifetime were measured in this manuscript to ascribe the luminescence nature from the solutions and pristine host materials at room temperature and upon cooling to 77K. This is important to reveal that the new host materials possess only fluorescence at room temperature.

The authors should provide the PL decay curves, IRF, fitted curves, and residual analyses.

Response: Agreed and added Figure S22 in Supporting Information.

The authors claim the thermal stability of NON-CF3. It should be supported by experimental results.

Response: The discussion of thermal stability in the manuscript is supported by thermal gravimetry experiments in the ESI Figure S12 for all materials.

Bandgap and/or T1 (, T2, T3..) energies of P170 should be compared with those of NON-CF3.

Response: Agreed and added phrase in the OLED section: “First two triplet states of **P170** are T₁ (³CT 2.77 eV) and T₂ (³LE 3.29 eV)³⁰ are lower in energy compared to first triplet state of the **NON-CF₃** host (T₁ is 3.3 eV) thus preventing Dexter energy transfer back to the host material.”

Why do EL spectra change with different host materials?

Response: We agree with referee that one might predict a red-shift of electroluminescence when increasing the host polarity for NON-CF3 molecule due to high permanent electric dipole moments of 7.9 D compared to much less 0.2 D for TCP and 1.4 D for mCP. However, this does not take in to account the solid-state interaction between host and guest – CMA molecules, which possess a large transition dipole. There are several interactions at play, but the largest is electrostatic. The guest has a large permanent dipole, such that dipole-dipole interactions are strongest where the host also has a permanent dipole, as is the case for NON-CF3. These interactions stabilize the guest’s ground state with respect to the excited state, leading to an increase in vertical excitation energy (an effect also observed in solution as negative absorption solvatochromism). In solid state, reorganization of the polarized matrix is suppressed, such that the blue shift is maintained also in emission. The mechanism behind this effect is discussed in more detail in our and others previous communications and tested across numerous hosts to demonstrate that CT emitters with large permanent dipole moment, allowing host–guest interactions as a tool to tune electroluminescence in OLED devices up to 200 meV EL shift.

Added text: “The phenomena of the PL^{30b,c} and EL^{30d} blue-shift of the CT emission for CMA materials was studied in detail and explained by thermally activated diffusion and electrostatic interactions between CMA emitter with host molecules.”

Minor

The MO diagrams are difficult to distinguish. Using a larger isodensity value is encouraged to regenerate the MO contours. Viewing from the top may also enhance visibility.

Response: We agree and the MO contour plots were updated in Table S4, showing three viewing angles front, side and top.

S is the common symbol for the overlap integral, not O.

Response: We believe that it's a community dependent abbreviation while we more commonly encounter symbol "O" for the overlap integral.

Table 1 annotation: $E_{\text{LUMO}} = E_{\text{opt gap}} - E_{\text{HOMO}}$ -> $E_{\text{LUMO}} = E_{\text{opt gap}} + E_{\text{HOMO}}$

Response: Agreed and corrected.

Figure 3, Table 2 and the corresponding texts do not match. i.e. temperature, PLQY of NON-Br at RT solution, etc..

i.e. Solvent, concentration should be included in the figure captions. The figures should stand alone.

Response: Agreed and added, while concentrations were added as a note "b" Table 1.

Energy diagram is required for the materials used in the OLED fabrication.

Response: Agreed. The requested diagram was added to Fig 4.

Reviewer #1 (Remarks to the Author):

All reviewers' comments have been addressed, and the paper is now ready for publication.

Response: We thank referee 1 for the expert opinion.

Reviewer #2 (Remarks to the Author):

The authors have addressed my points clearly. I have nothing further to add.

Response: We thank referee 2 for the expert opinion.

Reviewer #3 (Remarks to the Author):

The manuscript has been well revised according to the reviewers' previous comments. Only a few points can be discussed before publication in Communications Chemistry.

Previous comment: The authors should provide the PL decay curves, IRF, fitted curves, and residual analyses.

Response: Agreed and added Figure S22 in Supporting Information.

Table 2 and Fig S22: the RTPL decay curves are too close to the IRF. In that case, the calculated lifetime might be dominated by the IRF. Fitted curves and residual analyses with χ^2 is still missing.

Response: Agreed and added residual analysis tables S8 and S9 in ESI.

Previous comment: S is the common symbol for the overlap integral, not O.

Response: We believe that it's a community dependent abbreviation while we more commonly encounter symbol "O" for the overlap integral.

S (or S_{ij}) is the widely accepted and commonly used symbol for the overlap integral in quantum chemistry. O is sometimes used for ease of notation; however, S is the standard symbol for the overlap integral. I still recommend the authors use S to avoid confusion.

Response: Agreed and corrected.